# Dual Cross-Linked Starch–Borax Double Network Hydrogels with Tough and Self-Healing Properties

**DOI:** 10.3390/foods11091315

**Published:** 2022-04-30

**Authors:** Xiaoyu Chen, Na Ji, Fang Li, Yang Qin, Yanfei Wang, Liu Xiong, Qingjie Sun

**Affiliations:** 1College of Food Science and Engineering, Qingdao Agricultural University, Qingdao 266109, China; 17854277165@163.com (X.C.); jn87163@163.com (N.J.); qinyjnu@163.com (Y.Q.); 201701026@qau.edu.cn (Y.W.); xiongliu821@163.com (L.X.); 2Department of Food, Yantai Nanshan University, Yantai 265700, China; 17852023306@163.com

**Keywords:** starch, double cross-linked network hydrogels, dynamic covalent bonds, self-healing

## Abstract

Herein, we have fabricated starch–borax double cross-linked network (DC) hydrogels with tough and self-healing properties using a one-pot method. The addition of borax significantly increased the storage modulus and loss modulus of these starch–borax DC hydrogels. The maximum compression stress (~288 kPa) of starch–borax DC hydrogels containing 5% borax was about ten times greater than that of a pure-starch hydrogel. The texture profile analysis values of the DC hydrogels—including hardness, springiness, cohesiveness, and adhesiveness—increased compared to pure-starch hydrogels. In addition, starch–borax DC hydrogels exhibited excellent self-healing and shape-recovery properties. These DC hydrogels, with a variety of excellent properties, have potential applications in agricultural, biomedical, and industrial fields.

## 1. Introduction

Hydrogels are three-dimensionally (3D) cross-linked polymeric networks formed by hydrophilic polymer chains [1,2,3], and they have wide applications in the fields of food [4], drug delivery [5], actuation [6], and tissue engineering [7]. Due to the nontoxicity and biocompatibility of natural polymers, natural hydrogels based on starch [8], alginate [9], cellulose [10], chitosan [11], carrageenan [12], and protein [13,14,15] have wide applications in food, biomaterials, drug delivery, etc. Starch, due to its low cost, abundance, and high cross-linking ability, is an ideal material for preparing hydrogels [16,17,18]. At low temperatures, amylose chains form double helices due to the physical cross-linking of the hydroxyl groups in each side chain, thereby forming a 3D hydrogel network [19,20]. Our previous study reported that corn starch can form physically cross-linked hydrogels via amylose hydrogen bonding [19]. However, these starch hydrogels exhibited weak mechanical properties because of the single network with its low-density physical cross-link [21]. In addition, their mechanical properties and structure were destroyed and were irreparable when the hydrogels suffered some cracks [22], which limits their lifespan, as well as their applications in agricultural and biotechnological industries [16,23]. Thus, enhancing the mechanical properties, as well as introducing self-healing into hydrogels, have emerged as important ways to improve their durability, reliability, and maintain performance.

At present, a strategy for preparing chemically and physically double cross-linked network hydrogels has been used to enhance mechanical properties with better self-healing efficiency [22]. Borax, as a dynamic covalent cross-linking agent, has been applied in multiple fields, such as agriculture [24], biomedicine [25,26], and water treatment [27]. It can form dynamic boron ester bonds with hydroxyl functional groups of polysaccharides and enhance the mechanical properties of hydrogels [28]. Spoljaric et al. (2014) demonstrated that the addition of borax could significantly enhance the malleability of poly (vinyl alcohol) hydrogels [29]. Liu et al. (2020) reported that fenugreek gum–borax hydrogels exhibited significant heat resistance, which was due to the reversible borate/didiol bond between fenugreek gum and borax [24]. Furthermore, the addition of borax can improve the mechanical properties of starch films. It was reported that the tensile strength of the high-amylose, maize-starch film increased by about four times with the addition of 6 wt% borax-modified nanocellulose [30]. Lu et al. reported that, compared with pure-starch film, the tensile strength of the starch film with 10% borax-cross-linked starch nanoparticles increased by about 45% [31]. Therefore, we hypothesized that the addition of borax could enhance the mechanical properties of starch hydrogels by forming physical hydrogen bonds and dynamic covalent boron ester bonds.

In the current study, we demonstrated a feasible one-pot method for creating starch–borax DC hydrogels with excellent mechanical properties. The DC hydrogel networks were constructed by forming a hydrogen bond between starch chains and a dynamic boron ester bond formed between the starch and the borax. The rheological, mechanical, microstructural, and self-healing properties of the DC hydrogels were systematically analyzed. With excellent mechanical behaviors and self-healing properties, we believe that the application of renewable and biocompatible starch hydrogels could be expanded in the fields of agriculture, food, biomedicine, and tissue engineering.

## 2. Materials and Methods

### 2.1. Materials

Borax was obtained from Tianjin Guangcheng Chemical reagent Co., Ltd. (Tianjin, China). Corn starch (amylose content: 28.0%) was obtained from Zhucheng Xingmao Corn Development Co., Ltd. (Weifang, China).

### 2.2. Preparation of Starch–Borax DC Hydrogels

Starch–borax DC hydrogels were prepared via one-pot method. Corn starch (4.2 g) was added to 23.8 mL distilled water with borax concentrations of 0%, 0.5%, 1.0%, 3%, and 5.0%. Afterward, the mixed slurries were heated for 30 min at 100 °C to ensure complete starch gelatinization. Subsequently, the obtained starch pastes were stored for 6 h at 4 °C to form starch–borax DC hydrogels.

### 2.3. Mechanical Properties

A texture analyzer with a P 0.5 probe (TA-XT plus, Stable Micro Systems, Surrey, UK) was used to characterize the mechanical properties of the hydrogel samples at room condition according to the study of Ge et al. [32]. The compressive stress–strain measurements were conducted at room temperature by using the P36R probe. The cylindrical hydrogel samples with about 20.0 mm in diameter and 6.0 mm in height were measured.

### 2.4. Rheological Measurements

Rheological measurements of hydrogels were performed using a DHR-Hybrid Rheometer (TA Instruments, Anton Paar, Graz, Austria) using a parallel plate system (PP50). The linear viscoelastic region (LVE) was determined before the shear measurements through a shear strain sweep test (0.01–1000%) at a constant frequency (1 Hz), at 25 °C, according to a previously described method, with some modifications [32].

To ensure the dynamic viscoelasticity, the starch pastes were equilibrated to 50 °C and subjected to frequency sweep experiments, employing angular frequency ranges from 0.1 to 100 rad/s a shear strain of 1.0%. The freshly prepared starch pastes were subsequently cooled to 4 °C for 6 h to obtain starch–borax DC hydrogels. For the same-frequency sweep test of the starch–borax DC hydrogels, the storage modulus (G′), loss modulus (G″), and loss tangent (tanδ = G″/G′) were obtained.

The time sweeps of DC hydrogels were performed by cooling the hydrogels to 4 °C at the platform and maintaining that temperature for 120 min, with a frequency of 1 Hz and a shear strain of 1%.

The amplitude oscillatory strains were performed according to a study by Huang et al. [25]. The DC hydrogels were cooled from 25 °C to 4 °C, with a speed of 4 °C/min. Subsequently, shear strains of 1.0% and 500% were applied every 5 min, alternately, at a fixed frequency (1 Hz).

### 2.5. Characterization

For the structural analysis, the starch hydrogels and starch–borax DC hydrogels were frozen at −80 °C, vacuum-lyophilized, and then ground to obtain the freeze-dried hydrogels. The X-ray diffraction (XRD) of the freeze-dried hydrogels was measured with Cu Kα radiation using an XRD diffractometer (AxSD8 Advance, Bruker, Karlsruhe, Germany). An FTIR spectrophotometer (NEXUS-760, Thermo Nicolet Corp., Madison, WI, USA) was used to record the spectra of freeze-dried hydrogels. Prior to analysis, the freeze-dried hydrogels were pressed into pellets with potassium bromide. The method was based on previous studies, with minor modifications [19]. The morphology of the hydrogels was evaluated using a scanning electron microscope (SEM, S-4800, Hitachi Instruments Ltd., Tokyo, Japan). The samples were snap-frozen and then lyophilized for 3 days. The freeze-dried hydrogels were coated with gold before being mounted onto the specimen stage to observe their morphologies.

### 2.6. Self-Healing Assay

The starch–borax_3.5%_ DC hydrogels were broken in half. Then, the cut interfaces were placed together, without any stimulus, at 25 °C. After that, the samples were stretched from both ends to confirm the healing ability. The method was based on previous studies, with minor modifications [33].

The starch–borax DC hydrogels were molded into diverse shapes. Then, the hydrogels were cut into smaller pieces and put into diverse shape molds. They were molded at 25 °C. The self-healing process was observed by taking photographs.

### 2.7. Statistical Analysis

All the experiments were implemented in triplicate. Statistical analysis was conducted with Tukey’s test using SPSS V.17 statistical software (SPSS Inc., Chicago, IL, US). A significance level of *p* < 0.05 was used.

## 3. Results and Discussion

### 3.1. Rheological Properties

The variation in storage modulus (G′), loss modulus (G″), and loss tangent (tan δ) for starch hydrogels with different concentrations of borax at 50 °C (as a function of frequency) are shown in Figure 1. The G′ values of the pure-starch hydrogel remained unchanged within the test range, which indicated that the pure-starch sample strongly exhibited typical gel behavior. In the test frequency range, the G′ for each sample was at least one order of magnitude greater than G″, indicating that the elastic gel network had been established (Figure 1A). At a low angular frequency (≤1.0 rad/s), the G′ value for the lower-borax (0.5%) sample was higher than that of the higher-borax (2–5%) sample. With increasing angular frequency (>2 rad/s), there was a significant increase in G′ values with a 5% addition of borax.

As Figure 1B shows, the starch–borax hydrogels exhibited a typical gel network, in which tanδ values are below 1. The tanδ values of the starch–borax hydrogels were increased compared to the pure-corn-starch hydrogel, which suggested that the starch–borax hydrogel structures were weaker than those of the pure-corn-starch hydrogels. These results demonstrate that the covalent interactions between starch and borax can modify the rheological behavior of starch hydrogels.

Figure 2 shows the G′, G″, and tanδ curves of the starch–borax DC hydrogels, with various levels of borax content, prepared at 4 °C for 6 h, with varying angular frequencies. In all the hydrogels, the G′ values were greater than the G″ values for all angular frequencies (Figure 2A), suggesting solid-like behavior. For pure-starch hydrogels, the tanδ increased with increasing angular frequency. However, the tanδ values of all the starch–borax DC hydrogels initially increased with increasing angular frequency (Figure 2B), reaching a peak of 4 rad/s, and then decreased to the region of 4–100 rad/s. These results demonstrate that the starch–borax DC hydrogels prepared at 4 °C for 6 h had solid-like characteristics.

Figure 3 shows the time sweep of the starch−borax DC hydrogels, upon cooling from 50 to 4 °C at a rate of 2 °C/min, and being maintained at 4 °C for 120 min. Upon cooling, the G′ and G″ values for all samples increased rapidly for 23 min. The values of G′ and G″ remained almost constant during the process of 4 °C for 120 min, indicating that starch–borax DC hydrogels were formed at 4 °C.

The G′ and G″ of the hydrogels, when the alternate step strain was switched from a small strain (γ = 1%) to a large strain (γ = 500%) at a fixed frequency (1 Hz), are presented in Figure 4. When the strain was returned to 1%, the values of G′ and G″ for the samples (starch, starch–borax_0.5%_, and starch–borax_2%_) could not return to their initial values (data not shown). These results confirmed that the network structure of corn starch hydrogels (starch, starch–borax_0.5%_, and starch–borax_2%_) were destroyed. However, as the strain γ was raised to 500%, the G′ and G″ values of the starch–borax DC hydrogels (starch–borax_3.5%_ and starch–borax_5%_) also decreased sharply, and the G′ values were higher than the G″ values. After 300 s, when the strain returned to 1%, G′ and G″ quickly returned to their initial values, which suggested the rapid recovery of the internal network and mechanical properties of hydrogels. After three cycles, the starch–borax DC hydrogels (starch–borax_3.5%_ and starch–borax_5%_) could return to their original state. The results showed that the starch–borax_3.5%_ and starch–borax_5%_ DC hydrogels returned to their starting state in 300 s after 500% shear strain for three cycles, confirming the recovery of the internal network of the starch–borax DC hydrogels. The excellent self-recovery ability of starch–borax DC hydrogels was likely because of the formation of dynamic boron ester bonds. Lu et al. (2017) found that microfibrillated cellulose-poly (vinyl alcohol)-borax hydrogels could form reversible covalent cross-links between the hydroxyl groups of poly (vinyl alcohol) or microfibrillated cellulose and borate ions, as well as enhance the elastic response and self-healing ability [34].

### 3.2. Textural Properties

The textural parameters of starch hydrogels and starch–borax DC hydrogels are shown in Table 1. Compared with pure-starch hydrogels, the hardness value of starch–borax DC hydrogels with 5% borax increased from 342 to 650 g. The reason for a significant increase, via starch–borax, in DC hydrogels’ hardness may be the formation of two interpenetrating networks: a hydrogen bond cross-linked starch chain network and a covalently cross-linked network between starch and borax [30,31]. These results suggest that the occurrence of starch–borax interaction reinforces the starch hydrogels.

With the concentration of borax from 0% to 5%, the springiness, cohesiveness, and adhesiveness of starch–borax DC hydrogels increased. In addition, the springiness values in the range of 0.857–0.931 suggest their high elasticity. The cohesiveness of the tested gel increased with the addition of borax, indicating stronger interactions between corn starch and borax. Zhang et al. (2015) demonstrated that tea polysaccharide has the ability to increase the cohesiveness of wheat starch gel via the interaction between tea polysaccharide and wheat starch [35].

Figure 5 illustrates the compressive stress–strain curves of the corn starch hydrogels and starch–borax DC hydrogels. The fracture stress of the pure-corn-starch hydrogels was lower compared to the starch–borax hydrogels. When the borax content was 5%, the fracture stress of the starch–borax DC hydrogel reached a maximum value of 288 kPa, which increased ten times compared to the corn starch hydrogel (29 kPa). When the borax content was 5%, the fracture strain of the starch–borax DC hydrogel was 94%, compared to the pure-corn-starch hydrogels at 62%. This was caused by the double cross-linking, including the hydrogen-bonded cross-linked starch network and the dynamic boron ester bonds. The results further demonstrated the formation of a temporary starch–borax cross-linked the second network via dynamic boron ester bonds because the dynamic network could serve as a sacrificial network to dissipate the stress generated during deformation. Zhang et al. (2018) reported that the poly(acrylamide)/poly(vinyl alcohol) hydrogel was strengthened by the formation of the poly(vinyl alcohol)-borax boronate ester bonds [36].

### 3.3. SEM Analysis

The SEM observations of corn starch hydrogels and starch–borax DC hydrogels are presented in Figure 6 at the same magnification. All the corn starch hydrogels and starch–borax DC hydrogels exhibited a porous interconnected network structure. As shown in Figure 6A, the corn starch hydrogel had an interconnected network in its internal morphology, and its pore sizes were 10−20 μm. With the addition of borax, the number of starch–borax DC hydrogel pores increased, indicating an increase in cross-linking density (Figure 6B–E). As the concentration of borax increased from 0.5% to 3.5%, the pore size of the starch–borax DC hydrogels became smaller, and the pore distribution was more uniform within the range of 5–30, 5–20, and 1–10 μm (Figure 6D), suggesting the formation of a more compact structure. This phenomenon may be due to the interaction between starch chains and borate ions, leading to a double network formation. It has been reported that the average pore size of the PVA/borax hydrogels decreased from 19.8 μm to 14.4 μm as the concentration of borax increased from 0.4% to 1% [37]. A further increase in borax concentration to 5.0% resulted in a decreased number of cavities with thick-walled structures (Figure 6E). This is possibly due to the increased cross-linking density of starch–borax DC hydrogels at higher concentrations of borax. Li et al. (2019) found that borax–guar gum hydrogel had uniform interconnected macroporous structures created through boron ester bonds and hydrogen bonds [22].

### 3.4. X-ray Diffraction (XRD)

XRD patterns of hydrogels are presented in Figure 7A. The pure-corn-starch hydrogel exhibited B-type crystalline structures, with peaks at 17°, 18°, 22°, and 24°, which correspond to crystalline planes (031), (211), (231), and (132), respectively [38]. This was due to amylopectin recrystallization by forming a double-helical structure. However, with the concentration of borax from 0% to 5%, the X-ray patterns of the starch–borax DC hydrogels changed from B-type to amorphous structures. The phenomenon indicated that corn starch and borax interacted, thereby blocking the formation of hydrogen bonds in starch chains and preventing amylopectin from creating the crystalline starch region. The interaction between the hydroxyl groups of corn starch and the borate ions of borax could prevent starch chain alignment and suppress the retrogradation of amylopectin, resulting in the formation of an amorphous structure. Sethi et al. (2020) reported that the interaction between xanthan gum and starch via covalent bonding can suppress starch recrystallization and form an amorphous structure [39]. Li, Liu, Chen et al. (2019) found that guar gum interacted with the borax, and its crystallinity decreased [22].

### 3.5. Fourier Transform Infrared (FTIR) Spectroscopy

The FTIR diffractograms of corn starch hydrogels and starch–borax DC hydrogels are presented in Figure 7B. The FTIR spectrum of corn starch hydrogels had a characteristic absorption peak at 3299 cm^−1^ because of the vibration of –OH. A peak at 2926 cm^−1^ was observed because of the vibration of C–H. The peaks at 1645, 1454, and 1148 cm^−1^ belonged to O–H bending vibration, C–H bending vibration, and C-O-C stretching vibration, respectively [40]. In the FTIR spectra of starch–borax DC hydrogels, with the increase in borax concentration, the peak was changed from 3299 cm^−1^ to 3278 cm^−1^. The red-shift phenomenon was caused by the consumption of the hydroxyl groups of corn starch chains to form boron ester bonds with borax. Lu et al. (2019) found that the decreased free hydroxyl groups in the borax cross-linked starch nanoparticles could be attributed to the formation of boron ester bonds via the interaction between starch and borax [31]. Thombare et al. (2017) found that the formation of borax cross-linked guar gum hydrogels was associated with the covalent bonds between galactomannan and borax [41]. Lv et al. (2019) reported that the O–H stretching peak of locust bean gum/gellan gum hydrogels shifted from 3384 to 3330 cm^−1^, which was ascribed to the borate ester bonds formed by locust bean gum chains and borax [26]. For starch–borax DC hydrogels, the intensities at 1645 cm^−1^ and 1148 cm^−1^ decreased, and the band near 1454 cm^−1^ nearly disappeared. The band shapes and intensities at 849, 759, and 704 cm^−1^ were changed after cross-linking via borate ions. The results indicated that the OH groups in the corn starch chains could be cross-linked via borax on the starch–borax DC hydrogels. Huang et al. (2019) found that the peak at 833 cm^−1^ was ascribed to the B−O stretching in B(OH)^4−^, demonstrating the presence of borax [25]. Spoljaric et al. (2014) demonstrated that poly(vinyl alcohol)-borax hydrogels exhibited peaks in the range of 1333–1423 cm^−1^, which was due to the formation of cross-links between hydroxyl groups in poly(vinyl alcohol) and borate ions in borax [29]. Thus, the FTIR results confirmed the interaction between starch and borax via the formation of boron ester bonds in the DC hydrogel’s network.

### 3.6. Self-Healing Behaviors

The self-healing behavior of starch–borax_3.5%_ DC hydrogels was studied, and the results are shown in Figure 8A. The cylindrical hydrogels (Figure 8(Aa)) were sliced into two halves (Figure 8(Ab)). The two pieces were placed back together without applying any external force. The hydrogels quickly merged into a single piece of hydrogel, and visually notable changes occurred at the interface (Figure 8(Ac)). After that, the hydrogel was stretched without breaking to demonstrate the excellent self-healing properties (Figure 8(Ad)). The starch hydrogel without borax did not exhibit self-healing characteristics (data not shown). The self-healing mechanism of starch–borax DC hydrogel is shown in Figure 1. The destruction and reconstruction of boron ester bonds are in dynamic balance [42,43]. When the two pieces of starch–borax DC hydrogel were put together, after being cut into two halves, the unassociated groups, as well as the groups after dissociation at the interface, could react with one another to form new boron ester bonds, and thus form a single piece of hydrogel.

The starch–borax DC hydrogels with excellent shape-recovery properties are shown in Figure 8B. The starch–borax DC hydrogels were molded into diverse shapes (the letter “J”, a flower shape, and a five-pointed star shape). After that, the hydrogels were divided into smaller pieces and placed in diverse shape molds at 25 °C for 1 h. The starch–borax DC hydrogels could return to their initial shapes (the letter “J”, a flower shape, and a five-pointed star shape). After two cycles, the original shapes were recovered, demonstrating good recovery performance. The dynamic borax ester bonds could lead to the full recovery of the original shapes.

## 4. Conclusions

In this paper, starch–borax DC hydrogels with excellent compression properties and self-recovery capacity were designed and synthesized. The compression strength of starch–borax DC hydrogels containing 5% borax reached 288 kPa at a 94% compression strain, which was increased 10-fold compared to pure-starch hydrogels. Furthermore, the starch–borax DC hydrogels showed remarkable self-healing properties at room temperature. Because of their simple and efficient preparation process, starch–borax_5%_ DC hydrogels may have wide applications in agriculture, biomedicine, materials, and other fields.

## Data Availability

The data that support the findings of this study are available on request from the corresponding authors. The data are not publicly available due to privacy or ethical restrictions.

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
