# Peer review of "Dual Cross-Linked Starch–Borax Double Network Hydrogels with Tough and Self-Healing Properties"

_foods, 2022, doi:10.3390/foods11091315_

Round 1
Reviewer 1 Report
This paper studies the effect of borax addition in the starch-borax double network hydrogel formation. The starch-borax hydrogels, compared to pure starch hydrogel, have excellent mechanical and self-healing properties with potential applications. The manuscript is interesting, although mayor revision should be considered.
The authors must be consistent with the nomenclature, what is exactly DC? In line 10: starch-borax double network (DC), in line 34: double-network (DC), the acronym must be stated just once. Revise the rest of the document.
- Line 15: the word ‘adhesiveness’ is duplicated.
- Line 49: microcosmic? What do you mean?
- Line 57-58: please, add more information about the chemicals.
- Lines 66-68: Why the authors chose a P 0.5 probe? TPA tests should have been performed by using a probe bigger that the sample (double compression test).
- Line 67: please add the temperature.
In general section 2.4. has to be revised and rewritten for a better understanding.
- Line 77-79: it is already explained in section 2.2, please remove. The temperature was 95 or 100ºC?
- Line 82: please add the shear strain.
- Lines 83-86: add the shear strain and join with the previous paragraph.
- Lines 87-91: rewrite the paragraph for a better understanding. For example: In line 87, the reference number is missing, there are two references for the same method and after that all the conditions are again explained. In line 90, authors should use ‘shear’ strain in all the document. In line 91: change ‘strain’ per ‘frequency’. Why authors use different frequency units? Please unify.
Section 2.6. The authors should specify that it was performed only in starch-borax3.5% samples.
Section 3.1. Rheological Properties.
From a rheological perspective, a true elastic gel network has been established when G′ is at least 1 order of magnitude greater than G″ and either modulus is not or is only slightly dependent on frequency (Morris et al 2012). Please consider this in your discussion.
- Line 160: add shear strain and frequency.
Figure 4. Change the first label of Starch-borax 5% G’’ per G’.
-Lines 188-191. Authors should add some references to confirm that mechanism.
Table 1. Statistical letters must be revised, for example: The Hardness in Starch-borax0.5% and Starch-borax3.5% samples have different letters (not possible).
-Lines 326-327: Remove the last sentence.
Conclusions: Authors should guide about the which formulation would be recommended.
Reviewer 2 Report
The manuscript Dual Cross-linked Starch/Borax double-network hydrogels with tough and self-healing properties demonstrates the efficiency of the combination of starch-borax to develop a functional hydrogel with self-healing properties. The manuscript is well designed, and the characterizations of as-prepared samples are sufficient which is performed using various techniques.
Introduction
-There are a previous studies related to starch-borax to develop a functional material, for instance:
- Lu, N. Ji, M. Li, Y. Wang, L. Xiong, L. Zhou, L. Qiu, X. Bian, C. Sun, Q. Sun, Preparation of borax cross-linked starch nanoparticles for improvement of mechanical properties of maize starch films, J. Agric. Food Chem. 67 (2019) 2916–2925.
- Zhai, S. Gao, Y. Xiang, A. Wang, Z. Li, B. Cui, W. Wang, Cationized high amylose maize starch films reinforced with borax cross-linked nanocellulose, Int. J. Biol. Macromol. 193 (2021) 1421–1429.
It might be interesting to comment on these studies to reinforce the initial hypothesis of the research.
-The main properties analysed were mechanical and self-healing properties of the developed material, for this reason a brief introduction to its advantages and main applications would complete this section.
Materials and methods
- Line 76 …method, with some modifications [31].
What kind of modifications have been made?
-Line 62: … Afterward, the mixed slurries were heated for 30 min at 100 °C and Line 79: ... and heated for 30 min at 95 °C in a water bath.
Why is there a difference of 5ºC during the processing of the materials analyzed and the mixtures that the viscosity is then determined?
-Line 114: …. Statistical analysis
What type of statistical test has been applied? For instancia, Tukey’s test, Duncan….
Results and discussion
- Line 229: …. pore distribution was more uniform
What is the value of the pore distribution? How does this distribution compare with that obtained by other authors in similar systems?
- Line 244: … B-type crystalline structures, with peaks at 17°, 18°, 22°, and 24° which was due to amylopectin recrystallization by forming a double-helical structure.
Which crystalline planes of diffraction correspond to the peaks found in XRD?
- Line 279: … The band shapes and intensities at 849, 759 and 704 cm−1 were changed after crosslinking via borate ions
These modifications are not clear in the proposed figure.
-Line 299: Self-Healing Behaviors
Related to self-healing behaviors, Have the properties of the formulated material been evaluated after the self-healing process? Different authors have been demonstrated that the properties of materials decrease significantly after this process. This is a critical point in the development of this line of research.
Reviewer 3 Report
Тhis manuscript is devoted to the study of dual cross- linked starch/borax hydrogels, which is a very interesting issue in particular in fields of agriculture, food science and biomedicine. The authors have very successfully managed to show the relevance of the topic by preparing a complete and focused literature review, which cites a number of studies in recent years.
The methodology is described correctly. The results are clear and well explained.
Based on the high quality of the manuscript I suggest it to be accepted after minor revision.
Line 72-76: What is the temperature at which the linear viscoleastic region rheological measurements were made?
Figure 2A: It is not clear which values are for G’ and which are for G’’
Round 2
Reviewer 1 Report
The manuscript was amended following the suggestions.